# The Internet, Apps, and the Anesthesiologist

**DOI:** 10.3390/healthcare11223000

**Published:** 2023-11-20

**Authors:** Samuel Smith, Andrew Houghton, Brydie Mockeridge, André van Zundert

**Affiliations:** 1Department of Intensive Care Medicine, Redcliffe Hospital, Brisbane, QLD 4020, Australia; 2Faculty of Medicine, University of Queensland, Brisbane, QLD 4072, Australia; andrew.houghton2@health.qld.gov.au (A.H.);; 3Department of Anesthesia and Peri-operative Medicine, Royal Brisbane and Women’s Hospital, Brisbane, QLD 4029, Australia; 4Department of Anesthesia, Mater Hospital, Brisbane, QLD 4101, Australia

**Keywords:** anesthesia, peri-operative medicine, mobile applications, digital health, mHealth

## Abstract

Modern anesthesia continues to be impacted in new and unforeseen ways by digital technology. Combining portability and versatility, mobile applications or “apps” provide a multitude of ways to enhance anesthetic and peri-operative care. Research suggests that the uptake of apps into anesthetic practice is becoming increasingly routine, especially amongst younger anesthetists brought up in the digital age. Despite this enthusiasm, there remains no consensus on how apps are safely and efficiently integrated into anesthetic practice. This review summarizes the most popular forms of app usage in anesthesia currently and explores the challenges and opportunities inherent in implementing app use in anesthesia, with an emphasis on a practical approach for the modern anesthetist.

## 1. Introduction

Medicine has embraced the Internet and mobile technologies at a rapid pace. Whilst the Internet initially relied on computers the size of cabinets, just 50 years later, pocket-sized smartphones have almost two million times their memory capacity and almost limitless potential [1]. These devices are ubiquitous, with 2022 ownership rates greater than 90% in many developed countries [1]. Thus, the cross-over of mobile technology into health was always inevitable. 

Using Internet-connected mobile devices such as smartphones, anesthetists have access to information in an expedient manner, up to date, and available almost anywhere, anytime, including at the bedside. There are many areas within anesthesia that could be aided by mobile technology: peri-operative risk assessment, drug formularies and calculators, clinical decision-making support, crisis cognitive aids, billing, as well as on-the-go education [2,3,4,5,6,7,8]. Given the prevalence of mobile devices, it should come as no surprise that smartphone software applications (or apps) have experienced considerable uptake into anesthesia. In both Australian and American studies, 88–97% of anesthetists reported using smartphone apps in their clinical practice [9,10]. There remain, however, many questions about the legal, ethical, and practical usage of mobile technology within anesthesia [11,12]. 

There are few guides assisting in choosing or using anesthesia-related apps. With hundreds of thousands of available medical apps, the choice of which to use can be overwhelming [13]. Furthermore, the lack of technology education in medical curricula may mean anesthetists are unaware of the utility of available apps and thus are missing out on a potentially valuable adjunct to their clinical practice. Choosing the correct mobile app has medico-legal, financial, and clinical ramifications, both for individual clinicians and institutions. This article aims to summarize the usage of apps within anesthesia, as well as the issues related to their usage. We then provide a framework to evaluate a medical app for potential use within anesthesia clinical practice. 

## 2. Apps Currently Used in Anesthesia

### 2.1. The Definition of an App and its Use in Anesthesia

Mobile applications (or apps as they will be referred to) are software programs downloadable from the Internet “app stores” (e.g., Apple App Store/Google Play) that provide functionality for mobile devices [14]. Apps may be designed by the device manufacturer, or they may be created by independent third-party software developers. A 2022 report estimated that up to 99.99% of available Apple apps were these so-called “third-party apps” [15]. Whilst devices usually come pre-loaded with essential apps, users can download more apps from an app store either free, for a one-off purchase fee, or with a rolling subscription. 

According to current estimates, there are well over 350,000 health-related applications available, and this number is expected to double every few years [16,17]. Whilst this number includes consumer-facing apps, apps have also found their way into mainstream medical practice, including anesthesia. Anesthesia-specific health apps can fulfill multiple roles to ensure optimal patient care [17,18]. These include clinical decision support, dosage calculators, and drug interaction guides, providing peri-operative risk and other relevant calculators, providing access to journals and textbooks, and facilitating advanced patient monitoring (Appendix A) [17,18,19]. A 2020 survey showed 97% of Australian anesthetists reported using smartphone applications in their work [10,20]. Similar to prior studies, the most common types of apps used were drug and medical references [10,18]. Figure 1 outlines one classification system of anesthesia apps using app functionality, informed by previous reviews and the WHO classification of digital health interventions [17,21,22,23]. 

One can also classify anesthetic-specific apps by their anesthetic area of use (Figure 2). This alternative method of classification has two functions: firstly, it helps illustrate the many anesthetic-specific functions of apps and secondly, it has utility in focusing search terms when looking for apps [17]. 

Not all app categories seen in Figure 1 are at the same implementation stage in clinical practice. Patient-centered apps and clinical communication apps are in their relative infancy and require slightly different considerations than clinical apps used by clinicians currently. Health record access apps are usually designed by the medical records system operator, and thus there is often little choice in which app one can use. Therefore, this article will focus on apps currently commonly used in clinical anesthetic practice, namely: clinical decision tools, drug references, medical calculators, and reference texts [24].

### 2.2. Point of Care Healthcare Provider Clinical Decision Support Apps

Point-of-care apps are apps designed to be utilized at the patient's bedside to guide clinical decision making, for example, UpToDate [25]. The key feature of these apps is that they convey information quickly enough to make a clinical decision, which differentiates them from simply an electronic textbook. In general, point-of-care apps provide succinct summaries of conditions or clinical scenarios and outline specific and actionable management options. To support this aim, point-of-care apps must have easy-to-navigate design and make heavy use of algorithms and images. Clinical decision apps are of particular use in emergent situations, where the burden of cognitive load is well documented. One example is Medical Assistance eXpert (MAX), which guides users through common anesthesia emergencies [5]. This app was shown to improve the technical skills of anesthetic residents in crisis scenarios by over 10% [5]. In addition to being a guide to the general management of a condition, point-of-care apps can also utilize patient-level data to guide decision making. This can vary in complexity from simple apps displaying an algorithm to follow, all the way through to continually transmitting patient-worn devices that provide AI-generated clinical advice [26]. Another anesthetic example is a visual reaction time app that assesses the depth of propofol sedation to alert to impending oversedation [8]. Other evolving point-of-care apps can automatically generate cardiac indices from a single snapshot of an arterial waveform or measure exact pupillary light reflex in unconscious patients [27,28]. The Point-of-care clinical support apps have been shown to improve skills in managing local anesthetic toxicity, malignant hyperthermia, and peri-operative blood prescribing [29,30,31,32]. This diversity makes point-of-care apps particularly useful to anesthetists who must be able to manage a broad array of problems across the physiological spectrum. 

### 2.3. Drug References and Medication Management Apps

Drug reference apps provide information about dosages, frequency, indications and contraindications, and adverse reactions to medications. Advanced features include pill identification (based on pill shape, color, or imprint), drug interaction generators, and automatic dose calculation. With an ever-increasing number of medications, it is almost impossible to stay abreast of every drug or formulation, especially as indications and safety advice change over time as a drug is in use. On average, 28 new medications are approved in the U.S. every year, each with its own interactions and anesthetic considerations [33]. Complicating this is the fact that 40% of patients over 75 years already present with polypharmacy, and this will only continue to grow as the population ages [34]. These patients are at high risk of medication-related harm; however, several apps are now available to help guide medication management in the peri-operative period, guiding which agents need to be discontinued, which anesthetic drugs should be avoided, which doses are required in specific patient groups (e.g., obesity, renal failure), and to help clinicians avoid medication-related harm [35].

There are several limitations to drug reference app use. These apps provide a very specific function within the larger clinical picture and cannot take over the role of the prescriber. Secondly, given the rapid pace of drug development, apps may not always be up to date, although they hold a large advantage over paper references in this respect. Finally, these apps suffer from regionality given geographical differences in drug availability. Thus, before using or purchasing a drug reference app, one must be sure it is applicable in one’s locality. 

### 2.4. Medical Calculator Apps

There are many mathematical formulae used daily in anesthesia, the most common being weight-based medication dosing. Whilst there may be something said for the sad state of arithmetic in modern medicine, there is undeniably a patient safety advantage in automating these procedures. Such apps have a wide array of uses. For example, one available app allows an anesthetist to calculate the carbon dioxide emissions and thus the environmental impact of inhalational anesthesia [6]. Another use of medical calculators is risk score calculation. Risk stratification scores have multiplied in recent years to support evidence-based decision-making, for example, the CHA_2_DS_2_-VASc of stroke in atrial fibrillation, the Child-Pugh score for cirrhosis grading, and the Apfel score for post-operative nausea and vomiting [36,37,38,39,40]. Having these scoring systems available on an app has several advantages. Firstly, it takes a relatively small amount of space for an app to store up to several thousand scoring systems, far more than any person’s memory could contain. Having such a large array of risk prediction tools on an app also means anesthetists can be exposed to scoring tools they may not otherwise have known about. Secondly, with improved access to these tools on an anesthetist’s phone, there is easier integration of evidence-based scoring systems into practice. Thirdly, most calculator apps provide clinical suggestions about how to proceed with management based on the risk scores produced. This saves time compared to a clinician calculating a score (which will take extra time without an app), and then using several resources to determine what to do with said score. Finally, increased availability of “big data” means more complex, but potentially more powerful, risk prediction tools can be produced that might only be able to be calculated using app technology. For example, the American College of Emergency Physicians COVID-19 management tool was rapidly developed using national datasets in response to the ongoing pandemic; however, it is complex and would be extremely difficult to do without an app. Dissemination and calculation via apps such as MDCalc allowed for rapid national and international uptake, improving management for patients [40]. The allure of medical calculator apps is the effective, rapid, conversion of patient parameters into actionable scores or numbers, allowing anesthetists to better provide evidence-based care [40]. 

### 2.5. Reference Texts/Journal and Education Apps

Reference apps centralize access to databases, journals, or books. As medicine evolves the ability to access the most up-to-date information and research on the go is invaluable as it affords the opportunity to essentially change practice to be in line with the best evidence in real-time. Apps may belong to either a single journal or provide a single login point to multiple journals or resources. With a reference app, a large array of multi-modal educational opportunities can be presented wherever an anesthetist is, without the need to carry voluminous texts around. Furthermore, as these apps are internet-connected, they can provide anesthetists with research updates on relevant topics, ensuring that one remains as current as possible. Apps with a pure education focus also exist, allowing mobile training in a wide array of areas, including simulated cases, general medical knowledge, and more. 

## 3. Potential Issues with Medical Application Use in Anesthesia

Despite the potential for medical apps in anesthesia, several issues may detract from their usage. These include regulation and the medico-legal aspects of app use, cybersecurity, and patient safety concerns. These issues are summarized below. 

### 3.1. Regulation and Medico-Legal Aspects of Apps

As discussed, over 99% of apps are developed by third parties not affiliated with major technology corporations. The degree of scrutiny these app developers are under is therefore minimal, and mainly limited to the technical functionality of an app, not the information. This is an obvious concern where an app may be used in clinical care; however, there is surprisingly little regulatory oversight in this area. The United States Food and Drug Administration (FDA) only applies rigorous scrutiny to apps that display or handle confidential patient data, or directly-control medical equipment (e.g., an insulin pump). Therefore, for most apps discussed in this article, there is essentially no mandatory oversight [16]. There are many medico-legal concerns with app usage including patient privacy, data security, information accuracy, and resultant harm, with no case testing exactly where liability lies [41]. Some bodies have begun providing guidance for app developers; however, this advice will always be region-specific and need constant updating as legislation changes. Integrating independent fact checking and using information in conjunction with clinical acumen thus remains imperative for safe app usage.

### 3.2. Cyber-Security

Cyber-security has two main dimensions to consider concerning mobile health apps. Firstly, apps that involve accessing confidential patient data (e.g., iEMR-linked apps) pose a direct risk of compromising patient information. Up to 92% of all smoking cessation apps in one study transmitted data to a third party (mainly for advertising), often unbeknownst to users [42]. The second risk is to the practitioner using the app. Unscrupulous apps, or even legitimate apps hacked by nefarious third parties, can be used to gain access to a device. To mitigate these risks, routine cybersecurity measures such as using complex passwords or a password manager, not sharing personal information, monitoring for data leaks, and avoiding using or sending patient or financial data should be followed. 

### 3.3. Patient Safety

The most important issue to consider with any new medical technology is patient safety. In recent studies, most anesthetists believed smartphones improved safety in theatre, however, these views vary. Interestingly, 30% more younger anesthetists (<3 years of practice) believed smartphones improved patient safety when compared to older anesthetists (>30 years of practice) [10]. Potential mechanisms for harm include inaccurate information, distraction, or direct pathogen transmission.

Information accuracy on apps ranks as a chief concern among physicians. In a 2020 review, there were 52 case reports of safety events related to app use, although most of these were theoretical (e.g., apps potentially over-diagnosing conditions), with no reports of serious harm occurring [43,44,45]. One of the major causes of these events was a lack of expert involvement in app development. In one review of urology apps, 20% of apps had no healthcare professional involvement at all [46]. This had clear flow-on effects with regard to accurate information. Whilst lack of clinician involvement was far more common in patient (rather than physician-facing) apps, this lack of expert involvement remains a significant concern.

Given apps are used on mobile phones, distraction is another patient safety concern. Up to 33% of anesthetic residents believe that smartphone use was perceived as distracting by other operating theatre staff [47]. However, in an Australian study, most anesthetists thought app use helped their practice overall. [48]. Other proposed methods of indirect harm include pathogen transmission; in a 2020 study, whilst most surveyed anesthetists believed smartphones were a source of healthcare-acquired infection (HAI), only 11.5% cleaned their phones daily, and 10% never cleaned their phones [49]. Clearly, between the potential benefits and direct and indirect potentials to harm, the implementation of smartphone apps into anesthesia may be more complex than it seems. 

### 3.4. Barriers to Implementation

Multiple practical issues may serve as obstacles to widespread app usage like usability, cost, and service factors. A poorly designed app with a difficult user interface is much less likely to be adopted, especially in high-stress, high-stakes scenarios, as can be encountered in anesthesia. The usability of an app’s interface is often linked directly to cost. Whilst many apps are free or have minimal cost, many higher-quality apps have significant subscription fees. When one considers downloading multiple apps, these costs can accumulate rapidly, and again, can be the difference between widespread adoption and obscurity. Finally, there are several institutional barriers to uptake. This begins in medical school, where there is limited training in digital health. With limited training, anesthetists can be hesitant in choosing and using an app in their practice. Additionally, limited support for integrating apps into the workflow by institutions, limited financial support from anesthetic departments, and variable physician confidence in choosing and using apps can all act as barriers to their routine implementation [16]. 

## 4. Evaluating A Medical App

Given the potential benefits of using mobile apps in medical practice, the question becomes how to choose an app that maximizes benefits whilst maintaining patient and provider safety. Figure 3 outlines our suggested approach for evaluating a medical app. To provide greater clarity, a worked example is provided in Appendix A. 

### 4.1. Identifying Needs

Given the number of apps available, it is easy to mindlessly download every new medical app recommended. Thus, the first step in efficient app usage is a needs analysis, asking “What aspects of my practice could be augmented by mobile app technology [50]?” Other methods of deciding what areas could be improved with a mobile app include asking:What aspects of anesthesia could I improve, or do I struggle to keep up to date on?What aspects of my practice are currently the most time consuming?Are there emergency algorithms I could have on my phone for reference?What aspects of anesthetic practice do I come across infrequently?What aspects of my anesthetic practice involve mathematics that could be done electronically?I have heard about a new app from others in the department; does this sound like something I need?

By identifying key areas that would benefit from adding in a medical app, the total amount and cost of unnecessary apps can be avoided, as well as avoiding security risks from unwanted apps.

### 4.2. Research Available Apps

Once a gap has been identified, the next step would be to identify potential apps that can fill this. The first port of call would be to simply use a conventional search engine and/or the App Store of choice (e.g., Apple App/Google Play Store) to find available apps using relevant keywords. By researching the app with a search engine as well as the App Store, one can also find reviews from previous users and research the organization that has developed the app. Many medical journals, e.g., the British Journal of Anesthesia, also include app reviews and there are even websites entirely devoted to evidence-based reviews of medical apps, such as iMedical Apps, the National Health Service (NHS) App library, and XCertiA reviews [50,51]. Another avenue for researching potential apps is discussing with colleagues. There is little evidence to confirm how app usage spreads amongst physicians; however, anecdotally word-of-mouth is a common route of dissemination. Useful sources would include anesthetists who have a subspecialty in the area of interest (e.g., pediatric anesthesia) or a colleague with experience utilizing smartphone technology in anesthesia. This latter group is important to consider because it will likely include more junior colleagues who may be otherwise overlooked [10]. 

In Figure 3, we have included bi-directional arrows going from steps 1 to 2 and vice versa. This is because word-of-mouth is a common method of spreading awareness of a new app; however, a recommendation does not mean one should download it immediately—there still needs to be thought given to a needs analysis to weigh up the pros and cons of downloading a new app.

### 4.3. Evaluate Using Rubric and Cross-Reference Information

Once a potential app/s has been identified it can be trialed in a non-clinical setting. We suggest using an evaluation rubric to help guide decision-making about introducing the app into clinical practice (Table 1). We developed this rubric as most existing medical app evaluation tools are research-focused, and perhaps too comprehensive for everyday use, as well as being generalist and not anesthesia-specific. Whilst our rubric utilizes key aspects from these validated scoring systems, it differs in several ways. Firstly, it has been made more specific for clinician-facing, anesthetic apps by dropping several less-relevant criteria from previous systems (e.g., ease of patient use) [13,52,53,54]. Secondly, rather than a comprehensive, research-focused tool that provides an average score for an app, our rubric uses a simplified Yes/No dichotomy for each item. This allows users to identify what features an app has (and does not have) and quickly evaluate the criteria that are relevant to their needs. Finally, some modifications have been made based on the author’s own experiences working with apps in anesthesia. 

Our rubric primarily evaluates three domains: (1) design/usability; (2) clinical content/supporting evidence; and (3) technological/cybersecurity. The design domain asks questions about the usability of the app. Previous guidance suggests apps should be simple and easy to learn, with consistent graphics and layout [13]. Additionally, we suggest that emergency information has a built-in shortcut, as this is time-critical information. The second criterion evaluates the quality of clinical information by testing credibility, applicability, and evidence based. Credibility is a form of screening test; if an easily verifiable piece of information is inaccurate then one must carefully consider whether this app can be trusted at all. The next question is more nuanced; if the clinical information in the app seems credible, is it applicable to local guidelines? Many anesthetic units have policies and procedures e.g., patient-controlled analgesia protocols, or the use of sugammadex and other expensive medications. Before relying on an app, it is worth checking with local colleagues or department directors to see whether there are relevant policies to consider. One aspect that may help lend credibility is whether the app has been developed by a relevant authority (for example, the Australian Red Cross’s warfarin reversal app) and whether there are references provided [55]. The final domain is technical. As this is one area where clinicians may lack expertise, there should be a very low threshold to discuss this aspect with hospital information and technology (IT) staff or more tech-savvy colleagues. This is strongly recommended if the app requires you to upload identifiable patient data (for instance, access to iEMR or clinical photography). These apps should involve complex password creation, multifactor authentication, and an abundance of caution. Other technical aspects include whether the app can work in an offline environment as many theatre environments may have poor or no internet connection. Finally, one needs to consider the cost. Most medical apps are free, and the modal cost for non-free apps is less than one dollar [21]. However, some apps have both purchase and ongoing subscription costs, which may impact the decision to purchase and use an app, both personally and at the institutional level (which may involve purchasing multiple licenses for a department). 

**Table 1 healthcare-11-03000-t001:** Evaluation Rubric [50,53,56].

Domain		Question	Y/N	Examples of Issues
**Design & Usability**	1.	Is the app easy to start using?		Does the app: Have a long start-up process? Require users to create an account? Note this is context-specific- for apps accessing/uploading confidential data, a password-protected app is advisable; for other apps, however, long start-up processes may impede use.
	2.	Is the app’s interface easy to navigate?		Difficult-to-navigate menus make using an app frustrating and lessen clinical utility, especially in emergencies.
	3.	Do the app’s images have sufficient quality?		For example, a picture of an anaphylactic rash to compare to needs to have sufficient resolution and be in color to be useful clinically.
	4.	If the app has emergency functions, are these easy to find?		If the app provides emergency information, there should be a one-click function to access these, for example, drug dosages for MH or anaphylaxis, or ALS algorithms.
**Clinical Content &** **Evidence Base**	5.	Is the information seemingly credible?		Compare information on the app with other references or your own knowledge.
	6.	Is the app's clinical guidance concordant with local anesthetic practice?		If the information is at least credible, then check if it is applicable locally with: Comparison with pre-existing protocols/procedures. Known/common local/geographic variances (e.g., medically significant specific populations living in your catchment area, local antibiograms). Departmental consensus and standards. Also, consider whether the app uses the correct units.
	7.	Does the app give specific enough guidance?		Often to reduce liability, apps may not provide specific guidance (e.g., drug dosages). Assess whether the app gives enough guidance for your practice.
	8.	Has the app been developed by a reputable organization?/has there been clinician input?		Can you see who has developed this app? Can you see if there was physician involvement?
	9.	Does the app provide evidence or references?		There should be referencing of information, with links to original articles.
	10.	Has the app been validated in published studies, or had peer review?		Try and search the app using a web browser. Do any clinical studies or reviews come up, or does the app discuss its validation process?
**Technical& Cybersafety**	11.	Is the app compatible with your phone’s OS?		If available on your phone’s app store it will be, however not all apps are usable on all devices, or some functionality may occasionally be lost.
	12.	Does the app have sufficient phone security? ^1^		If sensitive data is being input, there should be a password-protection process. Carefully check the app’s privacy policy
	13.	Does the app need an internet connection to function?		If the app does not function at all without an internet connection (or does not have an offline option), it may not work in an OT environment ^2^.
	14.	How large is the app?		If the app is large (e.g., >50 MB), you may need to make room for the app by deleting files.
	15.	Does the app give too many/not enough notifications?		Too many notifications represent an annoyance, whilst urgent updates or breaking medical news may be desirable.
**Ongoing Use**	16.	Are there ongoing subscription costs?		Also, consider whether there is an existing hospital/university license to use the app.
	17.	Is there a way to contact designers for support?		There should be developer contact details in the event of app malfunction or to submit feedback.
	18.	Has the app been updated?		Generally, an update should occur at least every two years, if not more frequently [44].
	19.	How long will I need this app/when will I review it?		Apps may be downloaded for a specific case or list; if so, then these apps should be deleted after use. If an app is downloaded for longer-term use, then a plan for review is suggested.
**Overall markers of subjective quality**	20.	Would I recommend this app?		Would you suggest to another colleague they use this app?
	21.	Would I pay for this app?		And if so, how much would be appropriate?
	22.	What is my overall star rating from 1–5		
	23.	Has this improved anesthetic practice?		

Y/N: Yes/No depending on feature present or not present. MH: Malignant hyperthermia; ALS: Advanced life support; OS: operating system; OT: Operating theatre; MB: megabyte. ^1^ If inputting clinical/personal information, does the app require a password +/− two-step authentication ^2^ This can be tested by placing the mobile device into flight mode and seeing if the app functions.

### 4.4. Preliminary Selection

After considering the evaluation, one can select an app with reasonable confidence to use in clinical practice. It is important to plan when you will re-evaluate the app. We suggest free apps between 3 and 6 months, and subscription apps before the next payment is due (e.g., monthly, or annually). 

### 4.5. Use in Clinical Practice

When ready to use the selected app in anesthetic practice, it is important to try and keep a written or at least mental record of the experience pre- and post-using the app, evaluating what changes have been made to your practice. It is also useful to confer with colleagues to see how they have integrated the app into their practice. Anecdotally, an app that may not seem of use can take on a whole new role if one integrates it differently, which sometimes occurs only after consultation with colleagues with more app experience. 

### 4.6. Re-Evaluate Apps

Re-evaluating apps is essential to discard unneeded applications. Clearing these apps is desirable for several reasons. Firstly, all apps are a potential portal for security breaches, especially if an app is discontinued and software is not updated. Minimizing the number of unnecessary apps is therefore an important security measure. Secondly, if an app charges a subscription, this will be charged regardless of usage, resulting in unnecessary costs. Thirdly, many medical apps require large amounts of storage space, and deleting unused apps frees this. There are also aesthetic and usability considerations, as multiple apps can make navigating a smart device difficult. If one is satisfied with the app, then continue to use it, re-evaluating as needed. If you find you are not, then discard the app.

## 5. Further Steps

The use of mobile applications in anesthesia is only likely to grow with time. This will have consequences for the current/future workforce grappling with the logistics of adopting this technology. Whilst personal responsibility in evaluating specific apps is a necessary and positive step, it will take a multifaceted and multi-agency approach to improve the safe acceptance of apps into routine anesthesia. The major issues needing to be addressed include app regulation and medicolegal protection, improving the mobile app marketplace to bring the right app to the right person improving app accuracy and quality control, and establishing departmental and specialty college standards and pathways to safe app use in anesthesia. 

Before considering these issues, one may wonder, given the obstacles to overcome, why persist or encourage app usage in the clinical setting? Such an approach, however, ignores the growing body of evidence that certain apps used judiciously can improve clinical practice and care of patients. Thus, attempting to curtail app usage is not likely to be helpful or effective, and is clearly not in line with current trends. In a recent cross-sectional study of medical students, over 80% believed that smartphone-based learning should be introduced into medical studies [57]. Thus, we must consider how apps can be safely implemented into clinical practice. Gagnon et al. described four main influencing factors in technology adoption in healthcare: technology-related factors (e.g., ease of use), individual factors (familiarity with technology), environment (patient/interprofessional impacts of technology), and organizational (e.g., leadership/requisite training) [58]. As no one group can effectively address all these factors, we believe a concerted, coordinated effort from app developers, device manufacturers, software engineers, regulators, clinicians, and health services is the most effective way to introduce apps safely and efficiently into clinical practice (Figure 4). 

### 5.1. Role of Device Manufacturers and App Developers

Device manufacturers are responsible for app stores, and thus which apps can be downloaded. One area the manufacturer can improve on is the definition of a “health app”, which currently does not have a commercial definition. By creating a definition and then certification for apps designed for use in healthcare, with basic standards attached, the general quality of health apps could be improved. Whilst manufacturers and app developers are not likely to certify information in apps and take on a regulatory role, they could control which apps appear in medical app searches, and ensure these apps have some basic developmental and technological standards. Examples of baseline standards could include demonstrating appropriate clinician input, referenced information, and an undertaking to regularly update the app, or else risk app removal. Introducing just a few measures for health apps would likely improve performance whilst at the same time being realistic for companies to enforce [44]. Additionally, for apps designed for healthcare professionals, app stores could limit purchasing to clinicians with formal identification. This low-level certification must be balanced with preserving app innovation, however. We also recommend improving search functionality in App Stores. Current search filters and store design are predominantly focused on upselling apps, not necessarily finding the best app; in previous studies, as little as 5% of results on app stores were related to the actual medical search terms used [59]. Improved search functionality and “health-app” certification could thus improve the uptake of mHealth apps available today. 

Software and app developers also have a role to play in improving the quality of mHealth apps themselves. This can be done through better needs analyses before developing apps, improved fact-checking, and health professional engagement, as well as a commitment to regular updates. Currently, much of the digital health industry is led not by a “needs-based” approach, but by what is easy to develop [50]. This results in both duplicate function apps, and large app-less gaps in clinician needs. In creating new apps, clinician engagement would likely improve app quality and usability. Sourcing feedback from a wide array of sources has become easier with the advent of dedicated clinical digital health bodies such as the Australian Digital Health Agency. Clinician engagement and testing would also be important in developing and reviewing content prior to release. As discussed above, a disturbing number of purported health apps currently have limited to no clinician involvement. Remedying this would involve breaking down silos between IT and clinicians, again improving overall app quality. 

### 5.2. Role of Clinicians, Health Services and Regulatory Agencies

From clinicians, we strongly recommend seeking out opportunities to be involved in the app development process, from development through to departmental implementation. The only way to create apps meeting anesthetic needs is to have anesthetists involved in developing and then reviewing apps. Clinicians also need to be involved in actively lobbying hospitals and professional bodies to create safe policies and procedures for app use in clinical practice. Currently, there is limited guidance as to how apps should be used in anesthetic practice. Without clinician input, the default of most health services is to fall back to an exclusionary approach. This article demonstrates the need for effective app use in the anesthetic practice. Clinicians must engage with policymakers to ensure guidelines balance safety with the practicality and realities of modern medical practice. Additionally, clinicians must provide feedback to app developers when there is inaccurate information or suboptimal design.

Finally, health services and government agencies must develop safe usage guidelines for applications, as well as clear frameworks outlining medicolegal safety for practitioners, recognizing that smartphone apps are increasingly becoming a part of standard practice. From professional bodies, we suggest committees review apps and create lists of suggested apps, and potentially development of their own apps if a need is identified. This is not without precedent; the Australian and New Zealand College of Anaesthetists (ANZCA) already list recommended anesthetic mHealth apps and the ANZCA Faculty of Pain Medicine have developed their own application for opioid conversions. Similarly, the American Society for Regional Anesthesia (ASRA) pre-operative anti-platelet app dramatically was released with rapid uptake, recording over 30,000 downloads in the first month alone [60]. Colleges and anesthetic departments also need to create policies that emphasize safe, evidence-based app usage, to enable practitioners to feel safe and supported, when using mHealth apps within guidelines. Finally, there may be a role for regulatory agencies to decide which apps can be used in clinical practice in some circumstances. Additionally, providing clear medico-legal guidelines to app usage will likely remove litigation uncertainty in the minds of potential users. Such guidance will be most effective in conjunction with clinicians and app developers, encouraging multi-level and closed-loop engagement around this issue. 

## 6. Conclusions

Apps in anesthesia have already been shown to have a multitude of roles. These roles extend from clinical communication to peri-operative risk score calculators, with new apps being developed every day. Given the rapid pace of this development, not all anesthetists are yet suitably equipped to safely integrate apps into their practice in the most time- and cost-efficient way. The background and evaluation method presented in this article provide a framework for anesthetists to incorporate apps into their clinical practice. For this technology to really meet its potential, software and device developers, clinicians, hospital services, and professional bodies and regulators must come together and decide on needs and user frameworks to ensure patients are getting the best care possible with new technology.

## Figures and Tables

**Figure 1 healthcare-11-03000-f001:**
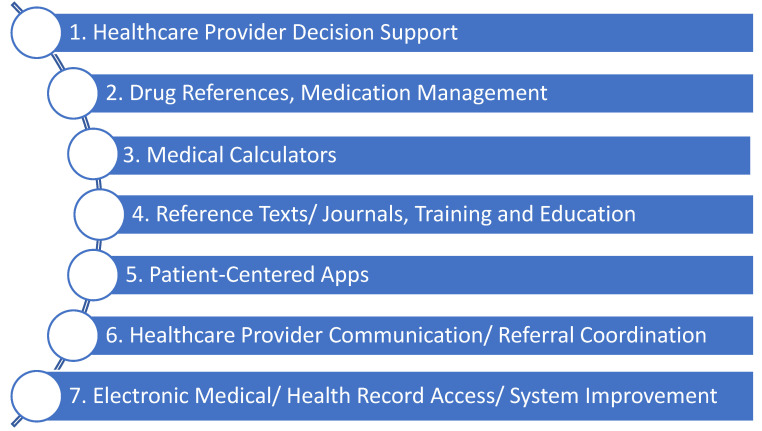
Functionality-based classification of anesthetic-related commercially available medical apps.

**Figure 2 healthcare-11-03000-f002:**
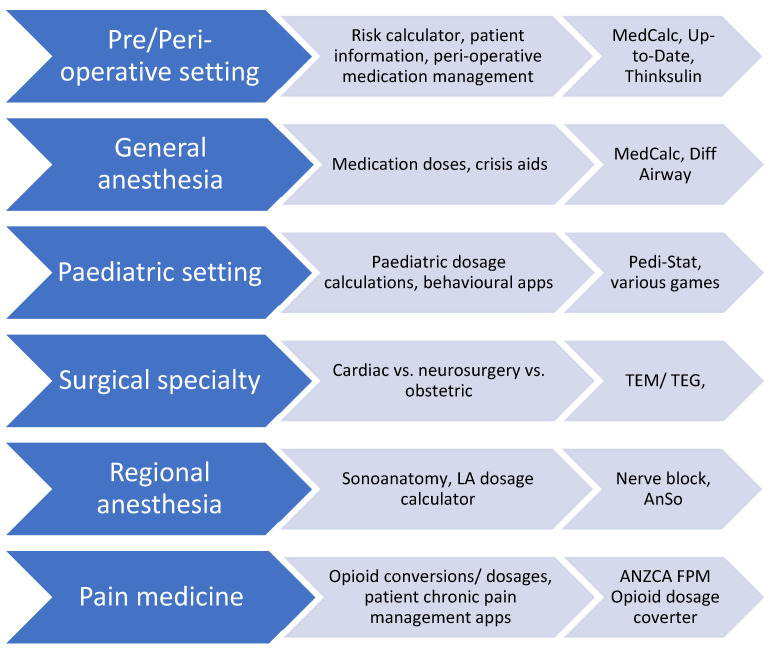
Specialty-based classification system of anesthetic applications.

**Figure 3 healthcare-11-03000-f003:**
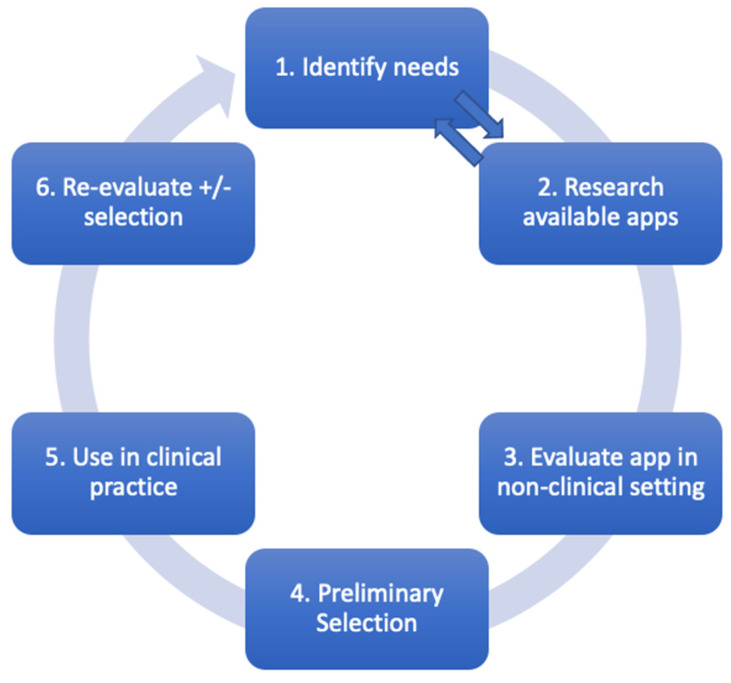
Proposed evaluation cycle for anesthesia app usage.

**Figure 4 healthcare-11-03000-f004:**
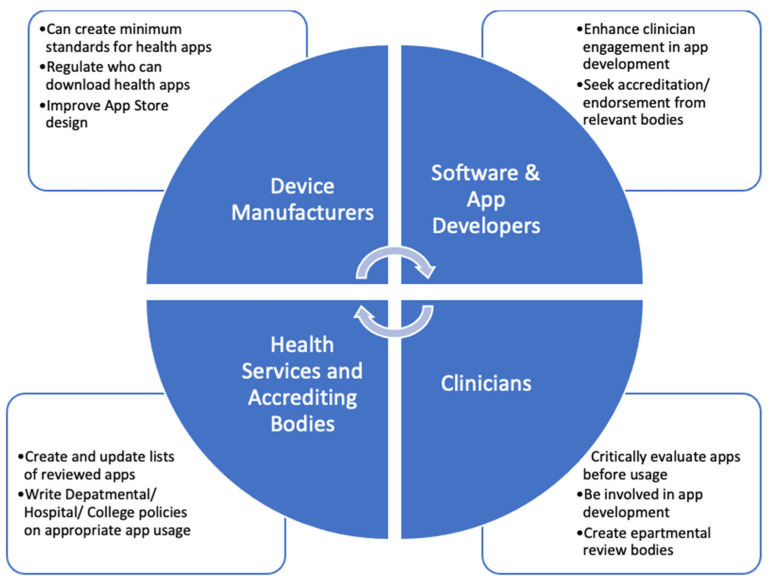
The interplay between different stakeholders in adopting health technologies and potential actions each group can take.

## Data Availability

No new data were created or analyzed in this study. Data sharing is not applicable to this article.

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
