# Peer review of "The Internet, Apps, and the Anesthesiologist"

_healthcare, 2023, doi:10.3390/healthcare11223000_

Round 1
Reviewer 1 Report
Comments and Suggestions for Authors
The purpose of this paper, a review of the current state of anesthesia apps and their potential for improvement to better align with medical criteria, is highly compelling. The analysis involves examining various apps and categorizing them based on their functionality and their applicability in the field of anesthesia. Their usefulness in decision-making processes, such as medication control, has been empirically established. However, this review also identifies limitations related to information updates, medical calculators, and accessing professional or academic documentation.
The paper also addresses the diverse challenges faced by these software programs, offering a comprehensive perspective on their limitations. Furthermore, it underscores that a significant barrier exists within the university system itself, marked by inadequate training in digital competencies.
A proposed evaluation cycle for the use of anesthesia applications is presented, emphasizing the pivotal role of medical specialists in aspects such as design, development, and usability. This approach underscores that involving anesthesiologists in both the development and subsequent review of applications is the only means to create software that truly meets the needs of anesthesia practitioners.
The manuscript is well-structured, transparently presenting the various facets of this subject matter. As a recommendation for future research endeavors, conducting a Delphi survey among anesthesiologists to compare the most commonly used apps and validate their relevance and reliability among observers using a validated questionnaire (such as DISCERN or GQS, among others) would be valuable. Additionally, a SWOT analysis could have also provided valuable insights.
Congratulations on producing such an original piece of work that is well-suited to the evolving landscape of eHealth.
Author Response
Thank you for your kind comments. We agree with the suggestions and points provided, and agree that for research, a Delphi approach would be valuable.
Reviewer 2 Report
Comments and Suggestions for Authors
Thank you for the opportunity to review the manuscript titled "The Internet, Apps and the Anesthetist". This article summarised the usage of apps within anesthesia, as well as the issues related to their usage, and provided a framework to evaluate a medical app for potential use within anesthesia clinical practice. Finally, it emphasized that software and device developers, clinicians, hospital services and professional bodies and regulators must come together and decide on needs and user frameworks to ensure patients are getting the best care possible with new technology. This is a very interesting review.
Author Response
Thank you for your kind comments.
Reviewer 3 Report
Comments and Suggestions for Authors
Hello, I am pleased to review this paper. The topic is important and the manuscript is written very well and on time. I believe it will be very useful for anesthetists, clinicians, educators in medical schools, heads of departments, Deans, researchers, app developers, mobile phone developers, as well as policymakers. It seems that the authors covered all aspects of the topic. Also, the supplementary file with a table and figure is a big plus.
I just have a comment on the following sentences.
"Device manufacturers are responsible for app stores, which apps can be downloaded and who can download them." The second part of the sentence after comma is not clear to me, is it true? Can't anyone download the apps?
"One area manufacturers can improve on is the definition of a “health app”, which currently does not exist. By creating a definition and then certification for apps designed for use in healthcare, with basic standards attached, the general quality of health apps could be improved."
The word "certification" should be used carefully in my opinion. If manufacturers have the power to certify the apps, they may limit the app developers in some sense, and limit the 'freedom' of app development. This may lead to 'certified' and 'not certified' apps with some advantages for some developers. So, I am not sure that manufacturers should be the body of certification. Instead, the manufacturers may validate (?) the apps. Perhaps, you may find a more appropriate word like validation, verification, etc.
Also, one minor issue with the names of the figures.
Please check the figure name in the supplementary file. Shouldn't it be Figure S1 as written on page 6 'Supplementary Figure 1'?
In the manuscript, you have the following mismatch:
Line - 133 Figure S2” Worked Example
This should be Figure S1 ”Worked Example".
Also, there is a typo here: Figure 5. "epartmental" should be departmental.
Thanks again for very good work.
Author Response
- "Device manufacturers are responsible for app stores, which apps can be downloaded and who can download them." The second part of the sentence after comma is not clear to me, is it true? Can't anyone download the a
Thank you for this comment. We have corrected this to say that app usage is typically limited by the app developer, not manufacturer, i.e. most apps are downloadable by anyone, but the app developer may have credentialling for actual use. We have also noted in the manuscript however, that limiting app download is entirely feasible on the manufacturer end as well.
- "One area manufacturers can improve on is the definition of a “health app”, which currently does not exist. By creating a definition and then certification for apps designed for use in healthcare, with basic standards attached, the general quality of health apps could be improved."
The word "certification" should be used carefully in my opinion. If manufacturers have the power to certify the apps, they may limit the app developers in some sense, and limit the 'freedom' of app development. This may lead to 'certified' and 'not certified' apps with some advantages for some developers. So, I am not sure that manufacturers should be the body of certification. Instead, the manufacturers may validate (?) the apps. Perhaps, you may find a more appropriate word like validation, verification, etc.
We agree with the concern around a dichotomy between “certified” and “uncertified” apps. Whilst in healthcare we should demand rigorous scrutiny, one of the aspects that has made apps so versatile is developer freedom. We have clarified the statement to reflect that potential certification relates more to whether apps have met an acceptable standard in their development rather than validating the actual content. Whilst it is important for apps developers to have freedom in their development, we suggest that if a healthcare app doesn’t meet some degree of basic standards (e.g. physician involvement, use of some validated references if applicable), then it may not be suitable for use in patient care. This need not be rigorous, and we are sure app developers do not wish to take on the mantle of a regulatory body, but ensuring apps listed as being “healthcare-related” meet some basic standards before upload, some degree of patient safety can be obtained. We have also made specific comment as the reviewer has suggested, urging caution before stifling developer innovation.
Please check the figure name in the supplementary file. Shouldn't it be Figure S1 as written on page 6 'Supplementary Figure 1'?
Thank you, this has been modified.
In the manuscript, you have the following mismatch:; Line - 133 Figure S2” Worked Example; This should be Figure S1 ”Worked Example".; Also, there is a typo here: Figure 5. "epartmental" should be departmental.
Thank you, these have been modified.